# Youth Distance Running and Lower Extremity Injury: A Systematic Review

**DOI:** 10.3390/ijerph18147542

**Published:** 2021-07-15

**Authors:** Tatiana Paz, Rachel N. Meyers, Cayla N. Faverio, Yuxuan Wang, Emily M. Vosburg, Derek J. Clewley

**Affiliations:** Division of Physical Therapy, Department of Orthopedics, Duke University School of Medicine, Durham, NC 27710, USA; rachel.n.meyers@duke.edu (R.N.M.); cayla.faverio@duke.edu (C.N.F.); yuxuan.wang1@duke.edu (Y.W.); emily.vosburg@duke.edu (E.M.V.); derek.clewley@duke.edu (D.J.C.)

**Keywords:** cross country, distance running, youth, adolescent, lower extremity injury

## Abstract

Distance running is a popular youth sport. This systematic review identified and examined the effects of youth distance running and lower extremity musculoskeletal injury. Scientific databases were searched from database inception to May 2020. Two hundred and fifty-eight full texts were screened, with nine articles retained for data extraction. Seven of the studies were case reports or case series. Two case reports involved an apophyseal hip fracture. No correlation was found between running mileage or gender and sustaining an injury. Middle school runners reported fewer injuries than high school runners. Cross country accounted for less than 10% of injuries in youth under 15 compared to other track activities. The main finding was a paucity of research. Available literature suggests youth can participate in distance running with minimal adverse effects. One exception may be increased vulnerability to growth plate injury. Additional research is needed, especially in those under 10, as literature in this population is nonexistent.

## 1. Introduction

Distance running has become increasingly popular among youth athletes over the past decade. In 2007, an estimated 12 million children between ages 6 and 17 participated in running [1]. Distance running was the most common and second most common physical activity among girls and boys ages 12 to 15, respectively [2]. The participation of youth runners in long distance events has also progressed, with reports of youth marathon finishers as young as 7 years old [3] and 100-km ultramarathon finishers as young as 12 years old [4]. The increase of distance running can result in a surge of youth injuries, as one study demonstrated a 34% increase in running-related injury incidence in children six to 18 presenting to U.S. emergency departments from 1994 to 2007 [2].

Distance running-related injuries in the adult [5,6,7,8] and high school [9,10,11,12] populations are well researched, as opposed to injuries in youth runners, which led us to focus only on the middle school and younger population. Skeletally immature runners are different than adult counterparts as they may be more vulnerable to injuries involving the physis and muscular-tendon attachment sites [1,13]. Although there is insufficient evidence on the effects of youth distance running, runners younger than 15 have completed marathons with few adverse outcomes. From 1982 to 2007, 310 youth runners, ages seven to 17, finished the Twin Cities Marathon with less injury incidence than adult finishers [3]. Youth runners had a medical encounter incidence of 12.9 of 1000 finishers, compared to 24.6 of 1000 finishers in adults [3]. Of the 310 youth runners, only four (1.29%) required medical attention, all mild in severity, and required no intervention besides a short period of rest [3]. Those four athletes were between the ages of 16–17 and no runners younger than 15 required medical assistance. Likewise, the Students Run LA Program, from 1989–2018, had more than 63,000 youth runners, as young as 12 years old, complete a marathon with no reports of adverse outcomes [14]. However, a study of 225,344 children (ages 6–18) who presented to U.S. emergency departments showed that the highest injury rate (45.8 per 100,000 US population) was in runners between ages 12 and 14 [2].

Controversy exists whether distance running is safe for youth runners. The first statement on risks in distance running for youth was published in 1982 by the American Academy of Pediatrics (AAP), which disapproved long-distance running events for children prior to physical maturation [15]. The International Amateur Athletic Federation of England guidelines stated that intense training in children can cause physeal damage and unnecessary psychological stress [16]. The AAP statement was updated in 1990, recommending that until further data are available on the relative risk of endurance running, if children enjoy the activity and are asymptomatic, there is no reason to preclude them from training for and participating in endurance running events [17]. The latest updates, although based on expert opinion, recommend accounting for the maturity level of the runner [14]. Self-motivated youth runners should complete a supervised training program, remain pain and injury free, meet appropriate weight and height gains, and maintain adequate sleep and nutritional needs for the demands of a growing body [14].

To the best of our knowledge, there has been no systematic review to date published that has investigated the effects of distance running for those under the age of 15 and lower extremity musculoskeletal injury. Therefore, the purpose of this systematic review was to identify and examine all of the available literature specific to lower extremity musculoskeletal injuries in youth runners under the age of 15.

## 2. Methods

### 2.1. Data Sources and Searches

We followed the Preferred Reporting Items for Systematic Reviews and Meta-Analyses (PRISMA) guidelines. The study was registered in PROSPERO; registration number CRD42019136428. The search was conducted by a medical librarian in MEDLINE (via PubMed), EMBASE (via Elsevier), and Scopus (via Elsevier) using both keywords and subject headings representing distance running, running injuries, and the pediatric population. Editorials, commentaries, and animal studies were excluded. The search covered the time frame from database inception through 7 May 2020. Reproducible search strategies can be found in the Appendix A.

### 2.2. Subjects

Studies were eligible for inclusion if the following criteria were met: (1) participants were under 15 years of age, (2) the distance reported was at least 800 m, and (3) the athlete sustained an injury to the lower extremity from running. The exclusion criteria included: (1) sports that were not strictly running (i.e., soccer), and (2) study design that was an editorial, review article, conference report, or letter to the editors.

### 2.3. Operational Definitions

Distance running: Our operational definition for distance running was either described as cross country or a minimum of 800 m if the distance was specified. We chose 800 m as the minimum distance to be included in this review as this has been considered a distance when the body transitions from anaerobic to aerobic system utilization [18]. Furthermore, we wanted to include a distance that would be sensitive enough to yield the highest number of studies.

Youth running: Our operational definition of youth running was population under 15 years of age. Our focus was to maximize the potential to include runners of middle school age or younger and to exclude high school runners.

### 2.4. Study Selection

Results from our search strategy were uploaded into Covidence (Veritas Health Innovation; www.covidence.org), a systematic review software. Duplicate citations were automatically identified and removed by Covidence. Two reviewers (TP, RM) independently screened references by title and abstract. A third independent reviewer (CF) settled any disputes between reviewers.

### 2.5. Data Extraction

Two independent reviewers (TP, RM) extracted the data from the included studies. The data extracted included the study design, participant demographics, total running mileage, and type of lower extremity injury sustained.

### 2.6. Quality Assessment

Risk of bias was assessed using the Modified Downs and Black checklist [19] for non-case reports. The 26-item checklist consists of four subscales with the categories of reporting, external validity, internal validity (bias), and internal validity (confounding selection bias). Two reviewers (CF and YW) completed the checklist independently. A third reviewer (TP) settled disagreements between the two reviewers.

## 3. Results

### 3.1. Study Selection

The search strategy (Appendix A) resulted in 4384 articles. After duplicates were removed, there were a total of 2176 articles. Upon completion of the title and abstract screening, 258 articles were retained for full text screening. After full-text screening, eight articles were retained for data extraction. One additional article [20] was later identified through a hand search resulting in a total of nine articles that met eligibility criteria and were included in the qualitative analysis (Figure 1).

### 3.2. Study Characteristics

Two of the nine articles included were cohort studies: a retrospective descriptive epidemiological study and an observational prospective cohort study. Table 1 includes data extracted from those two studies. Seven of the nine included articles were case reports or case series. Data extracted from those articles are included in Table 2.

#### 3.2.1. Cohort Studies

The retrospective descriptive epidemiological study by Reid et al. gathered data from the National Electronic Injury Surveillance System (NEISS) of the US Consumer Product Safety Commission [21]. The study focused on runners aged 10 to 18 participating in various track activities and events from 1991 to 2008 [21]. The study reported an estimated 25,237 injuries in the 10 to 12 year old age group and 53,498 in the 13 to 14 year old age group [21]. Injuries that occurred as a result of participation in cross country accounted for 4.9% of the track-related injuries in the 10–12 age group and 9.3% of the injuries in the 13–14 age group [21].

The prospective study by Goldman et al. had a sample of 1927 students from 50 high schools and 34 middle schools that participated in the 2017–2018 Students Run Los Angeles (SRLA) Marathon Training Program [20]. There were a total of 720 seventh and eighth grade middle school participants and 102 of the 720 reported an injury [20]. Middle school runners were less likely to report an injury than high school runners (14.2% vs. 20.8%, *p* < 0.001) [20]. Middle school runners who sustained an injury over the training program ran significantly greater distance on average per week than non-injured runners (14.1 miles vs. 12.0 miles, *p* < 0.001) [20]. The most commonly reported injury site was the knee, followed by the lower leg, foot, ankle, thigh, and hip [20].

#### 3.2.2. Case Reports and Case Series

The ages in the seven case reports and case series ranged from 11 to 14. Four out of the seven runners were 14-year-olds. The cases reports and case series consisted of five males and two females. Training distances were up to more than 10 km per day [22]. All of the cases reported overuse injuries and symptoms persisted for an extended period of time prior to seeking medical care. There were three femoral injuries [22,23,24], two hip injuries [25,26], one tibial injury [27], and one patellar injury [28].

The femoral injuries included adductor insertion avulsion syndrome [22], bilateral supracondylar stress fractures [23], and a pathologic fracture [24]. Two of the three athletes (67%) that experienced femoral injuries [22,24] identified as members of a track athletics club. Both of the hip injuries [25,26] were apophyseal fractures and occurred in 14-year-old runners. The tibial injury was a stress fracture in an 11-year-old runner [27] and the patellar injury was a dorsal defect in a 13-year-old competitive runner [28]. For more details on the specific injuries, please refer to Table 2.

### 3.3. Risk of Bias

The last column of Table 1 summarizes the results of the risk of bias assessment using the Modified Downs and Black Checklist [19]. Total scores ranged from 15 to 17. The following total scores ranges on the Downs and Black Checklist have been suggested as categories of study quality: excellent (26–28), good (20–25), fair (15–19), and poor (≤14) [29].

## 4. Discussion

The purpose of this systematic review was to identify and examine the effects of youth distance running (age under 15) and lower extremity musculoskeletal injury. We found a scarcity of research with the majority of studies being case reports or case series. As such, we could not establish a conclusive relationship between youth distance running and lower extremity injury. Our systematic review provides limited information related to lower extremity injuries in youth distance runners and limited indications, suggesting distance running is safe for youth under 15.

The retrospective descriptive epidemiological study [21] provides some insight into the prevalence of lower extremity injuries in youth runners. Less than 10% of track-related injuries in the 10–12 and 13–14 age groups occurred during cross country events, meriting further discussion on if there is a significant correlation between youth distance running and lower extremity injury rates [21]. Goldman et al. also found Students Run Los Angeles runners had a high marathon completion rate with no injuries or adverse outcomes reported, also meriting further discussion that distance running could be safe for youth [20].

### 4.1. Type of Injury

From our findings, it is difficult to draw definitive conclusions of the most common injury type in distance runners younger than 15. All of the injuries experienced were overuse injuries and runners often had persistent symptoms prior to diagnosis. This is consistent with injury literature that suggests that prior injury is a risk factor for future injury. Early treatment of symptoms should be considered to avoid future injury and more serious pathologies. The observational prospective cohort study [20] found knee injuries to be most common among runners participating in the Students Run Los Angeles marathon training program. Overuse injuries are not unique to the pediatric population, as many adult runners also experience overuse injuries.

Many youth athletes have an increased susceptibility to stress-related growth plate injuries, particularly long distance runners [30]. Among the case reports in our systematic review, we identified two [25,26] out of the four (50%) 14-year-old runners experienced a growth plate injury to the hip, specifically an apophyseal fracture. Although youth are more vulnerable to growth plate injuries, especially around the age of 14, none of the injuries we identified in youth runners involved the epiphyseal-physeal-metaphyseal (EPM) complex. Youth runners are still at risk for growth plate injuries and are therefore reason for concern [2,13,30,31]; however, stress-related apophyseal injuries cannot adversely affect growth like stress-related periphyseal injuries. This is important to note as stress-related apophyseal injuries tend to not be treated long-term or directly affect the growth centers [30,32], suggesting that youth runners may not be at increased risk for irreversible growth disturbance or long-term deformity.

Our systematic review suggests there is emerging evidence that youth under 15 can successfully participate in distance running with relatively low injury rates. Our findings suggest distance running may have minimal adverse effects on youth runners. One exception may be increased vulnerability to growth plate injury.

### 4.2. Runner’s Gender and Injury 

Out of the seven case reports, we identified that all of the femoral injuries occurred in males. However, only two of the case studies were female, making it difficult to identify if there is a correlation between gender and injury location. Goldman et al. also found there were no significant differences (*p* = 0.153) between injury rates in male and female youth runners [20]. From our findings, there does not seem to be a definitive association between injury location and gender of the youth runner.

### 4.3. Comparison to High School and Adult Runners 

A majority of the studies in this review were excluded because we could not extract data from under 15 for the studies that had a range of ages inclusive of 18 and under, as the data was not broken down respective to age. Even though some of our studies did include 15 to 18-year-olds we could specifically extract individual data from under 15.

In the study by Goldman et al. [20], high school and middle school runners participated in a 28-week marathon training program with 14.2% of middle school runners reporting injury compared to 20.8% of high school runners [20]. Additionally, the study found that adolescent distance runners had a lower injury prevalence compared to adults during marathon training [20]. In 2018, there was also a 99% completion rate of Students Run Los Angeles runners in the Los Angeles Marathon with no reports of adverse outcomes [20]. These conclusions suggest that distance running may be safer for youth runners under 15 compared to high school and adult runners. In another retrospective cohort study [3], the race records demonstrated that of the 310 youth runners completing the Twin Cities Marathon, there were four medical encounters and none of these encounters occurred in runners under 15 years of age. This again suggests that distance running may be safe for youth runners, specifically runners under 15. On the other hand, of children and adolescents ages 6–18 years old treated in the emergency room, children ages 12–14 had the highest running-related injury rate of 45.8 injuries per 100,000 US population [33]. It should also be noted that there is inherent risk to participating in any type of running and these injury rates could reflect these risks. The reported injury rates could be expected from sport participation and not dependent on distance.

### 4.4. Growth and Development

Another important aspect to consider is that physiologic differences in youth can differ regardless of age. Age at specific developmental stages can also vary considerably between males and females because females, on average, physiologically develop earlier than males [4,14,34]. Additionally, during development children experience growth spurts, which could increase vulnerability for injury due to change in bone mineral density and peak height velocity [14,34]. Peak height velocity, the time of rapid growth during which bones are growing at a faster rate than muscles and tendons, typically occurs at age 12 in females and 14 in males [14]. All of the apophyseal fractures in the case reports and case series were in 14-year-olds, which is the typical age for adolescent growth spurts in males. Bone mineral content is also at its lowest right before peak height velocity [14]. During this period of growth and development, a child’s body may have decreased ability to withstand load. Proper management of these health aspects has the potential to help decrease the risk of growth plate injuries and stress fractures in youth runners.

### 4.5. Current Recommendations

While there is a scarcity of literature available on youth distance running, there are a considerable number of guidelines available to the public. The American Academy of Pediatric Council on Sports Medicine and Fitness (AAPCSMF) 2007 guidelines recommend limiting one sport to a maximum of five days per week with at least one day off from organized physical activity and two to three months off each year from that particular sport to allow for recovery and avoid burnout [1]. The National Athletic Trainers’ Association recommends pediatric athletes limit vigorous activity to 16 to 20 h per week [1]. The American College of Sports Medicine (ACSM) 2018 Physical Activity Guidelines for children and adolescents (ages six to 17) recommends participation in 60 min per day of moderate to vigorous physical activity [35]. The National Association for Sports and Physical Education (NASPE) also recommends children participate in at least 60 min of physical activity each day [36]. The International Amateur Athletic Federation recommended that athletes younger than 14 should not run more than 3000 m [31]. Furthermore, the Australian Sports Medicine Federation recommended that runners ages 9 and under, 9–11, and 12–14 run in a competition distance no more than 3 km, 5 km, and 10 km, respectively [37]. The Federation also recommended that youth under 14 should not run more than three times per week and that training sessions should not last longer than 90 min. Another study found that young athletes whose weekly training hours exceeded their age were more likely to experience an injury, especially an overuse injury, suggesting weekly training hours should be below an athlete’s age [38]. There is limited consistency among these recommendations, and they are predominantly based on expert opinion rather than evidence based. Taken together, the inconsistencies in current recommendations and the lack of literature revealed by the current review suggest that overuse injuries in youth runners and their association with distance and training time clearly needs more research.

Recommendations are not only created for training, but also for competitive events, which further emphasizes the need for guidelines substantiated from scientific evidence. The International Marathon Medical Directors position statement from 2003 recommended against participation in marathons for those under 18 [20]. The International Association of Athletics Federation (IAAF) implemented a minimum age requirement of 20 years to participate in the 2017 marathon’s World Championships in London [4]. The minimum age for participation in various marathons and races tends to be between 16 to 18 years of age. Furthermore, it is also common for a parent or guardian to sign a waiver giving permission for the young runner to participate [4]. On the other hand, the youngest athlete to compete in the 2016 Olympic Games in Rio de Janeiro was 13 years-old [4]. Minimum participation age is completely arbitrary, without any substantial evidence for the age selected. As noted in the results of our systematic review, additional research is needed on youth distance running to help race organizers create appropriate age participation requirements, if any at all.

### 4.6. Limitations

The limitation of this systematic review is that most of our studies were case reports, which inherently contain bias. Nevertheless, they provided qualitative information that allows for more insight into the types of lower extremity injuries experienced by youth distance runners. Due to the limited number of studies that met our search criteria, it is difficult to establish definitive conclusions or any association between youth distance running and lower extremity injury. This systematic review, however, established the need for more research in youth distance running. Additionally, this review demonstrated we do not have evidence to back recommendations for youth distance running, including what distances are safe and which injuries are most common to help reduce injury risk.

### 4.7. Implications

The implications of this systematic review are that, based on the available literature, an association cannot be drawn between long distance running and lower extremity injury in runners under 15. Mileage recommendations cannot be made for runners under age 15. There is not sufficient available literature to support if distance running increases lower extremity injury in runners under the age of 15. Although there is a paucity of literature, it is important to note that most of the available guidelines for youth distance running are based on expert opinion and not scientific evidence. This establishes the need for more research on the effects of long distance running on pre-pubescent runners. There was considerably more literature on high school runners, but the pre- and post-pubescent body has differences that can influence the ability to sustain increased mileage, withstand loads, and manage injury. Additionally, the youngest runner from the case reports was an 11-year-old and one cohort study included 10-year-old runners, establishing that there is a dearth of literature for those under 12 and literature for those under 10 is nonexistent. This suggests that we do not have any substantial scientific evidence to determine the effects of long distance running on this age group. It would be beneficial to have research specifically on pre-pubescent and skeletally immature runners to ensure appropriate recommendations for safe mileage, potentially leading to a reduction of injury rates in youth runners.

## 5. Conclusions

In this systematic review, we were able to identify and examine the literature specific to youth distance running and lower extremity musculoskeletal injury. The main finding of our study was that there was a dearth of literature related to injuries in runners under the age of 15. The available literature suggests distance running may be safe for runners under 15 and youth can participate in distance running with minimal adverse effects. This review helps establish the need for additional research on the effects of distance running on those under 15 to help develop evidence-based guidelines, mileage recommendations, and injury risk reduction protocols based on the most common injuries.

## Figures and Tables

**Figure 1 ijerph-18-07542-f001:**
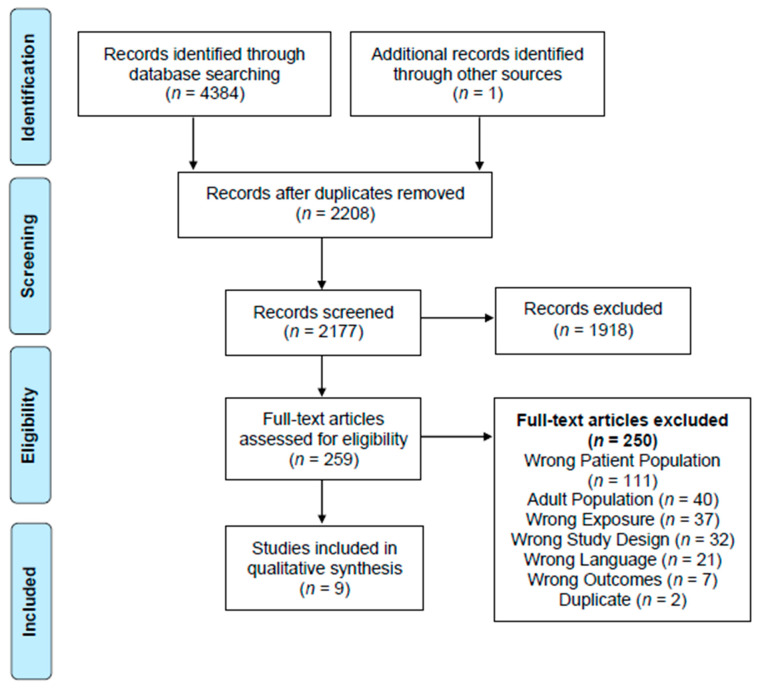
PRISMA Flow Diagram.

**Table 1 ijerph-18-07542-t001:** Quantitative Study Characteristics.

Author, Year, Design	Population (Sample Size, Age)	Running Exposure	Outcome	Downs and Black Risk of Bias
Cohort Studies				
Goldman et al., 2020 Observational Prospective Cohort Study	720 MS participants (7th and 8th graders) in SRLA marathon training program	28-week marathon training program 3 weekday training runs 1 long weekend run Began with 2-mile runs and increased to a maximum 20-mile run prior to marathon Week 5: 5K Week 14: half-marathon Week 9 and 17–20: school holiday closures Mean distance per week: 9.87–22.82 m	Injury sites across all runners Knee (33%) Lower leg (19%) Foot (14%) Ankle (13%) Thigh (6%) Hip (6%) 102 reported MS injuries HS runners more likely to report an injury than MS runners (*p* < 0.001) 20.8% of HS runners reported injuries versus 14.2% of MS runners (*p* < 0.001) MS runners who sustained an injury ran significantly greater distance on average per week than non-injured MS runners (14.1 mi vs. 11.5 mi, *p* < 0.001)	15
Reid et al., 2012 Retrospective Epidemiological Study	National Estimates *: Age 10–12: 25,243 injuries (95% CI: 20,125–30,362) Age 13–14: 53,504 injuries (95% CI: 43,286–63,722)	Cross country events	10–12 year-olds: Estimated 1234 (4.9%) injuries 13–14 year-olds: Estimated 4964 (9.3%) injuries Most frequently injured body part across all runners: Lower extremities (58.2%) Upper extremities (19.0%) Trunk (13.8%)	17

MS: Middle School; SRLA: Students Run Los Angeles; HS: High School * N was national estimates based on weighted data for 4496 actual cases from the National Electronic Injury Surveillance System of the US Consumer Product Safety Commission.

**Table 2 ijerph-18-07542-t002:** Case Report Characteristics.

Author, Year	Population	Running Exposure	Symptoms	Outcome
Case Reports				
Clancy et al., 1976	1 out of the 13 cases reported was a 14-year-old male cross-country runner	Cross country running	One-month history of gradual onset of pain around right anterior iliac crest Experienced pain when running and coughing or sneezing	Fracture separation of the anterior portion of right anterior iliac apophysis Localized tenderness over right anterior iliac crest. This pain was reproduced with resisted abduction of affected hip. Complete relief of symptoms and full return to running after four weeks of rest
Daffner et al., 1982	1 of the 4 cases reported was an 11-year-old male runner	Running 2–3 m per day	One-month history of pain and localized swelling in the proximal right tibia Pain with direct pressure to proximal tibia, running and walking	Right tibial stress fracture Mild swelling over proximal medial tibia 8 cm distal to joint line and radiograph confirmed circumferential area of periosteal new bone, thickened posteriorly Discharged on minimal activity and improved rapidly
Dull, 2000	14-year-old female competitive cross-country athlete	Running 20 to 30 miles per week including up and down hill running exercises	Bilateral hip pain localized to ASIS, anterior thigh and low back. Pain was greater on the left. Hill running seemed to exacerbate her pain more than any other activity. Initially tried conservative treatment and returned to a moderate running regimen, five days later pain intensified with sharp grabbing sensations in the anterior thigh and pelvis	Bilateral avulsion fractures of the pelvis apophyses Apophyseal separation fracture of the left anterior superior iliac crest Seven months after returning to activity following left apophyseal separation fracture, experienced avulsion fracture to right anterior superior iliac crest Training was ceased for a short period of time, and she returned to a successful running regimen within 12 weeks
Gamble, 1986	13-year-old female competitive athlete	Running, placed first in her age bracket in a 10 km race prior to onset of symptoms	Two-month history of increasing pain and swelling of right knee One-month prior to presentation experienced sensation of the knee giving way while trying to accelerate while running Tender to palpation of patella and discomfort with patellar compression and maximal quadriceps contraction	Symptomatic dorsal defect of the right patella Excisional biopsy was performed. Four months after surgery, she had full range of motion and no symptoms.
Nishio et al., 2012	14-year-old male member of track athletics club	Running more than 10 km daily	One-month history of progressively worsening pain in medial aspect of the left thigh Pain was initially experienced only after running but progressed to also occur with weightbearing activities	Adductor insertion avulsion syndrome CT: presence of periosteal reaction and intracortical linear hypoattenuation and showed no fracture line MRI: periosteal, cortical, and intramedullary signal intensity abnormalities Treated with initial avoidance of weight bearing using crutches for ambulation, followed by progressive weight bearing for two weeks. Symptoms resolved completely seven weeks after initial evaluation and he had normal gait without pain. At three months returned to gradual running program.
Ross et al., 2008	14-year-old male cross-country runner	Cross country training, began training 2 months prior to presentation	Three to four weeks into training developed pain in left distal thigh. Soon after also began to experience right thigh pain.	Bilateral supracondylar stress fracture Decreased activity to pain-free levels with acetaminophen for pain. At one-month follow-up he was pain free and allowed to gradually return to running.
Sakamoto et al., 2008	12-year-old male member of a track athletics club	Running more than 5 km a day	Seven-week history of discomfort in distal left thigh when running Symptoms were gradually worsening, with discomfort increasing to pain four weeks prior to presentation Tenderness over distal thigh and increased pain with weightbearing	Pathological insufficiency fracture associated with nonossifying fibroma in the distal femur CT: showed radiolucent cortical lesion seen on plain radiographs as a well-defined area of decreased density in the cortex, consistent with a diagnosis of NOF, with periosteal new bone Initial avoidance of weightbearing using two crutches for ambulation, followed by progressive weightbearing over five weeks. Three months after onset, he had normal gait without pain and returned to running.

ASIS: anterior superior iliac spine; Km: kilometers; CT: computed tomography; MRI: magnetic resonance imaging; NOF: nonossifying fibroma.

## Data Availability

Not applicable. Search strategy provided in Appendix A.

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
