# Peer review of "Youth Distance Running and Lower Extremity Injury: A Systematic Review"

_ijerph, 2021, doi:10.3390/ijerph18147542_

Round 1
Reviewer 1 Report
Dear Author,
I understand that you used PRISMA guidelines, but I not sure if we can called this review as systematic review if you decribe clinical cases in more cases in this review.
In my opinion exlusion and inclusion critieria are too poor.
In my opinion running 800 meters as a distance is not a typical distance running
Author Response
Dear Reviewer 1,
Thank you for your review of our manuscript. We have carefully reviewed the comments and have revised the manuscript accordingly. Our responses are given below in a comment-by-comment manner.
Sincerely,
Tatiana Paz, DPT
Comments and Suggestions for Authors:
Dear Author,
- I understand that you used PRISMA guidelines, but I not sure if we can called this review as systematic review if you decribe clinical cases in more cases in this review.
- With our systematic review we aimed to capture all available literature on youth running and lower extremity musculoskeletal injury. Due to the lack of available research, the majority of literature is comprised of clinical cases, but the full processes of a systematic review were followed, including PRIMSA guidelines.
- In my opinion exlusion and inclusion critieria are too poor.In my opinion running 800 meters as a distance is not a typical distance running
- 800 meters is classified as a middle-distance event. We chose 800 meters as the minimum distance to allow for a sensitive search that encompassed all available literature on youth running.
Reviewer 2 Report
This work has filled the gaps in the area of youth distance running under the age of 15 and muscle injury. Although the content of the manuscript is less formative, I agree that this manuscript may call for the attention of researchers to do more research on this issue.
Author Response
Dear Reviewer 2,
Thank you for your review of our manuscript. We have carefully reviewed the comments and have revised the manuscript accordingly. Our responses are given below in a comment-by-comment manner.
Sincerely,
Tatiana Paz, DPT
Comments and Suggestions for Authors:
- This work has filled the gaps in the area of youth distance running under the age of 15 and muscle injury. Although the content of the manuscript is less formative, I agree that this manuscript may call for the attention of researchers to do more research on this issue.
- Thank you, we appreciate your comment and noting our manuscript fills a gap in literature and calls for more research. Also, thank you for noting a minor spell check was required, we have corrected grammatical errors throughout the manuscript.
Reviewer 3 Report
Thank you for letting me review the manuscript entitled "Association of youth distance runing and lower extremity injury: a systematic review".
The background of the manuscript is quite clear. However, the design of a systematic review (with or without metaanalysis) should be used when there are a sufficient number of publications with suitable designs to allow a conclusion. The inclusion of 7 case reports or case series is not appropiate for a systematic review. More primary research must be done before this study.
Also, I would like to point out other considerations:
- References are not written correctly in the text
- The manuscript should be focus on the population of interest. There is too much information about other groups not included in the aim of the manuscript (in the results and discussion parts mainly).
- The types of injuries must be well-explained, otherwhise a classification cannot be done. In this case, the authors of the included studies did not provide the information. So, in my opinion, there is no information to carry out this systematic review.
- Discussion: "The purpose of this systematic review was to determine the association between 172 youth distance running (age under 15) and lower extremity injury." This is not the appropiate desing to determine an association.
- Due to the lack of information from the included studies, the discussion is a review of different subparagraphs that are not directly related with the aim of the study.
Author Response
Dear Reviewer 3,
Thank you for your review of our manuscript. We have carefully reviewed the comments and have revised the manuscript accordingly. Our responses are given below in a comment-by-comment manner.
Sincerely,
Tatiana Paz, DPT
Comments and Suggestions for Authors:
Thank you for letting me review the manuscript entitled "Association of youth distance runing and lower extremity injury: a systematic review".
- The background of the manuscript is quite clear. However, the design of a systematic review (with or without metaanalysis) should be used when there are a sufficient number of publications with suitable designs to allow a conclusion. The inclusion of 7 case reports or case series is not appropiate for a systematic review. More primary research must be done before this study.
- Thank you for your attentive feedback. We agree this systematic review would be more powerful if we were able to include more studies that were not published case reports or case series. Nevertheless, the process of a systematic review is still in place, including following PRISMA guidelines. We also wanted to ensure we captured all available literature on youth running and lower extremity musculoskeletal injury. We acknowledge this review resulted in a null finding, but null findings are still important in literature and encourage further research in the field.
Also, I would like to point out other considerations:
- References are not written correctly in the text
- Thank you for informing us of this error. We have corrected the references and the in-text citations utilizing the template provided.
- The manuscript should be focus on the population of interest. There is too much information about other groups not included in the aim of the manuscript (in the results and discussion parts mainly).
- The other populations mentioned in the results are high school runners and this population is mentioned as this was how the results were presented from the Goldman et al. study that included middle school and high school runners and compared their results. We elected to discuss the comparison between high school and adult runners in the discussion section because often the literature focuses on these populations, and we wanted to highlight the differences between these populations and youth runners under 15.
- The types of injuries must be well-explained, otherwhise a classification cannot be done. In this case, the authors of the included studies did not provide the information. So, in my opinion, there is no information to carry out this systematic review.
- In Tables 1 and 2 of the results section we go into further detail on each of the injuries, providing the extent of the detail we could gather from the included studies.
- Discussion: "The purpose of this systematic review was to determine the association between 172 youth distance running (age under 15) and lower extremity injury." This is not the appropiate desing to determine an association.
- This is a great point. We agree that we cannot draw an association from our systematic review. We have made adjustments in the language of our manuscript and adjusted the aim of the study so that is in more appropriate for our design. These changes can be noted in the title, abstract, discussion and conclusions on lines 1, 11, 12, 66-70, 192, 193 and 392-395.
- Due to the lack of information from the included studies, the discussion is a review of different subparagraphs that are not directly related with the aim of the study.
- In the discussion, we selected to include Growth and Development, Benefits of Running and Current Recommendations as subsections to provide background our particular population of focus. We chose to include the benefits of running to highlight what can be gained from running comparative to the low injury risk that was found in the studies we identified; this was also often a point of discussion in the background articles we found. Growth and development are very particular to the under 15 age group and is a key factor to consider when comparing runners of different ages. We also wanted to highlight that while research is limited there are a vast number of recommendations, indicating they are not based on evidence. We added content in line 323, 324, 336 to help illustrate this point.
Round 2
Reviewer 3 Report
I have no additional comments or suggestions.
Author Response
Dear Reviewer 3,
Thank you again for taking the time to review our manuscript. Thank you for letting us know that you have no additional comments or suggestions after our latest submission and making changes based on your feedback.
Sincerely,
Tatiana Paz, DPT